# International gestational age-specific centiles for blood pressure in pregnancy from the INTERGROWTH-21st Project in 8 countries: A longitudinal cohort study

Lauren J. Green[1]*, Stephen H. Kennedy[2,3], Lucy Mackillop[2], Stephen Gerry[4], Manorama Purwar[5], Eleonora Staines Urias[2], Leila Cheikh Ismail[2,6], Fernando Barros[7], Cesar Victora[8], Maria Carvalho[9], Eric Ohuma[2,4,10], Yasmin Jaffer[11], J. Alison Noble[12], Michael Gravett[13], Ruyan Pang[14], Ann Lambert[2], Enrico Bertino[15], Aris T. Papageorghiou[2,3], Cutberto Garza[16], Zulfiqar Bhutta[10ᴪ], José Villar[2,3ᴪ], Peter Watkinson[1ᴪ], for the International Fetal and Newborn Growth Consortium for the 21st Century (INTERGROWTH-21st)[2¶]

1 Critical Care Research Group, Nuffield Department of Clinical Neurosciences, University of Oxford, Oxford University Hospitals NHS Trust, NIHR Biomedical Research Centre, Oxford, United Kingdom, 2 Nuffield Department of Women's & Reproductive Health, University of Oxford, Oxford, United Kingdom, 3 Oxford Maternal & Perinatal Health Institute, Green Templeton College, University of Oxford, Oxford, United Kingdom, 4 Centre for Statistics in Medicine, Nuffield Department of Orthopaedics, Rheumatology & Musculoskeletal Sciences, University of Oxford, Oxford, United Kingdom, 5 Nagpur INTERGROWTH-21st Research Centre, Ketkar Hospital, Nagpur, India, 6 College of Health Sciences, University of Sharjah, United Arab Emirates, 7 Programa de Pós-Graduação em Saúde e Comportamento, Universidade Católica de Pelotas, Pelotas, Brazil, 8 Programa de Pós-Graduação em Epidemiologia, Universidade Federal de Pelotas, Pelotas, Brazil, 9 Faculty of Health Sciences, Aga Khan University, Nairobi, Kenya, 10 Centre for Global Child Health, Hospital for Sick Children, Toronto, Canada, 11 Department of Family & Community Health, Ministry of Health, Muscat, Sultanate of Oman, 12 Department of Engineering Science, University of Oxford, Oxford, United Kingdom, 13 Departments of Obstetrics and Gynecology and of Global Health, University of Washington, Seattle, Washington, United States of America, 14 School of Public Health, Peking University, Beijing, China, 15 Dipartimento di Scienze Pediatriche e dell' Adolescenza, SCDU Neonatologia, Universita di Torino, Torino, Italy, 16 Division of Nutritional Sciences, Cornell University, Ithaca, New York, United States of America

ᴪ These authors contributed equally to this work.
¶ Membership of the International Fetal and Newborn Growth Consortium for the 21st Century is provided in the Acknowledgments
* lauren.green@ouh.nhs.uk

## Abstract

### Background

Gestational hypertensive and acute hypotensive disorders are associated with maternal morbidity and mortality worldwide. However, physiological blood pressure changes in pregnancy are insufficiently defined. We describe blood pressure changes across healthy pregnancies from the International Fetal and Newborn Growth Consortium for the 21st Century (INTERGROWTH-21st) Fetal Growth Longitudinal Study (FGLS) to produce international, gestational age-specific, smoothed centiles (third, 10th, 50th, 90th, and 97th) for blood pressure.

### Methods and findings

Secondary analysis of a prospective, longitudinal, observational cohort study (2009 to 2016) was conducted across 8 diverse urban areas in Brazil, China, India, Italy, Kenya, Oman, the

**Data Availability Statement:** Blood pressures by gestational age underlying this work can be found at: https://github.com/StephenGerry/Blood-pressures-in-pregnancy Other relevant data are within the manuscript and supporting information files.

**Funding:** This study was supported by the Bill & Melinda Gates Foundation (SK, https://www.gatesfoundation.org/) and the NIHR Biomedical Research Centre, Oxford (PW, https://oxfordbrc.nihr.ac.uk/). The funders had no role in study design, data collection and analysis, decision to publish, or preparation of the manuscript.

**Competing interests:** I have read the journal's policy and the authors of this manuscript have the following competing interests: PW holds grants from the National Institute for Health Research; LM and AP are supported by the National Institute for Health Research (NIHR) Oxford Biomedical Research Centre (BRC). SG is funded by an NIHR Doctoral Research Fellowship (DRF-2016-09-073). The views expressed are those of the authors and not necessarily those of the NHS, the NIHR or the Department of Health and Social Care. LM works part-time for Sensyne Health. PW worked for Sensyne Health. Both have share options in Sensyne Health; ZB is a member of the Editorial Board of PLOS Medicine; no other relationships or activities that could appear to have influenced the submitted work.

**Abbreviations:** ALSPAC, Avon Longitudinal Study of Parents and Children; ANOVA, analysis of variance; BMI, body mass index; CRL, crown rump length; FGLS, Fetal Growth Longitudinal Study; GAMLSS, generalised additive models for location, scale, and shape; HELLP, haemolysis, elevated liver enzymes and low platelets; INTERGROWTH-21st, International Fetal and Newborn Growth Consortium for the 21st Century; ISSHP, International Society for Study of Hypertension in Pregnancy; LMP, last menstrual period; MAP, mean arterial pressure; MGRS, Multicentre Growth Reference Study; MOEWS, Modified Obstetric Early Warning Scores; SD, standard deviation; SSD, standardised site difference; STROBE, Strengthening the Reporting of Observational Studies in Epidemiology.

United Kingdom, and the United States of America. We enrolled healthy women at low risk of pregnancy complications. We measured blood pressure using standardised methodology and validated equipment at enrolment at <14 weeks, then every 5 ± 1 weeks until delivery.

We enrolled 4,607 (35%) women of 13,108 screened. The mean maternal age was 28·4 (standard deviation [SD] 3.9) years; 97% (4,204/4,321) of women were married or living with a partner, and 68% (2,955/4,321) were nulliparous. Their mean body mass index (BMI) was 23.3 (SD 3.0) kg/m². Systolic blood pressure was lowest at 12 weeks: Median was 111.5 (95% CI 111.3 to 111.8) mmHg, rising to a median maximum of 119.6 (95% CI 118.9 to 120.3) mmHg at 40 weeks' gestation, a difference of 8.1 (95% CI 7.4 to 8.8) mmHg. Median diastolic blood pressure decreased from 12 weeks: 69.1 (95% CI 68.9 to 69.3) mmHg to a minimum of 68.5 (95% CI 68.3 to 68.7) mmHg at 19$^{+5}$ weeks' gestation, a change of −0·6 (95% CI −0.8 to −0.4) mmHg. Diastolic blood pressure subsequently increased to a maximum of 76.3 (95% CI 75.9 to 76.8) mmHg at 40 weeks' gestation. Systolic blood pressure fell by >14 mmHg or diastolic blood pressure by >11 mmHg in fewer than 10% of women at any gestational age. Fewer than 10% of women increased their systolic blood pressure by >24 mmHg or diastolic blood pressure by >18 mmHg at any gestational age. The study's main limitations were the unavailability of prepregnancy blood pressure values and inability to explore circadian effects because time of day was not recorded for the blood pressure measurements.

### Conclusions

Our findings provide international, gestational age-specific centiles and limits of acceptable change to facilitate earlier recognition of deteriorating health in pregnant women. These centiles challenge the idea of a clinically significant midpregnancy drop in blood pressure.

## Author summary

### Why was this study done?

- Internationally applicable gestational age-specific centiles for blood pressure are needed in clinical practice to determine when women have left the "normal" range.

- It is uncertain whether clinically significant decreases in blood pressure occur between early and midpregnancy.

### What did the researchers do and find?

- We estimated international gestational age-specific blood pressure centiles using longitudinal blood pressure data provided by women from 8 countries who took part in the International Fetal and Newborn Growth Consortium for the 21st Century (INTERGROWTH-21st) Project.

- On average, systolic blood pressure rose by around 8 mmHg between 12 and 40 weeks' gestation, with no decrease in midpregnancy. Diastolic blood pressure decreased slightly

(by around 0.6 mmHg) between 12 and 19 weeks, rising thereafter until 40 weeks' gestation.

- At any gestational age, systolic blood pressure fell by >14 mmHg and diastolic blood pressure by >11 mmHg from baseline in fewer than 10% of women. Fewer than 10% of women increased their systolic blood pressure by >24 mmHg or diastolic blood pressure by >18 mmHg at any gestational age.

### What do these findings mean?

- Our findings challenge the frequently quoted midpregnancy blood pressure decrease, advocating for a higher index of clinical suspicion when a woman presents with a "lower than booking" blood pressure, especially in late pregnancy.

- We show the limits for acceptable change in blood pressure during healthy pregnancy, which should help clinicians determine patients with abnormal blood pressure rises and falls.

## Introduction

Monitoring blood pressure is a key part of antenatal care. Gestational hypertensive disorders result in over 70,000 maternal deaths annually worldwide [1]. However, physiological changes in blood pressure from early pregnancy onwards are not included in any contemporary definition. Instead, the International Society for Study of Hypertension in Pregnancy (ISSHP) uses a threshold to define gestational hypertensive disorders (including preeclampsia) as the new onset of systolic ≥140 mmHg or diastolic ≥90 mmHg blood pressure at or after 20 weeks' gestation [1]. There is no clear definition of hypotension in pregnancy [2], although it predicts evolving sepsis [3] and is associated with pulmonary embolism [4] and cardiac disease [5], leading causes of maternal mortality [5].

Not incorporating pregnancy-induced physiological changes in the definitions of gestational hyper- or hypotension is understandable. The extent of "normal" measures for each gestational age are not available. Obstetric textbooks [6,7] and e-learning packages [8] commonly report a midpregnancy dip of 10 to 15 mmHg for diastolic blood pressure. This would suggest that thresholds for hypotension should be lower in midpregnancy. However, this paradigm of blood pressure in pregnancy is derived from old studies with small patient numbers [9,10] or comparing mid-second trimester measures with those obtained in prepregnancy (which are often not available in routine clinical practice) rather than in the first trimester [11].

Larger studies including the Avon Longitudinal Study of Parents and Children (ALSPAC) suggest a smaller drop in mean arterial pressure (MAP) between the initial and midpregnancy blood pressures; however, those data are also dated and comprise routinely collected measures from a single geographical region [12].

To fill the knowledge gap for healthy pregnant women, free from major identifiable medical, nutritional, social, and environmental risk factors, with good maternal, perinatal, and child health outcomes, we describe blood pressure patterns from 12 weeks' gestation to delivery in a secondary analysis of the International Fetal and Newborn Growth Consortium for the 21st Century (INTERGROWTH-21st) Fetal Growth Longitudinal Study (FGLS) [13]. We

also determined whether site blood pressure data could be pooled to generate international gestational age-specific centiles for blood pressure to complement the international standards already published by the INTERGROWTH-21st Project [14].

Defining centiles for systolic and diastolic blood pressure across healthy pregnancies would help determine gestational hypertensive disorders, as well as hypotension thresholds beyond which conditions such as pregnancy-related sepsis and haemorrhage should be considered. These could be incorporated into blood pressure thresholds for Modified Obstetric Early Warning Scores (MOEWS).

## Methods

This work is reported following the Strengthening the Reporting of Observational Studies in Epidemiology (STROBE) guidelines [15] (S1 STROBE Checklist).

### Study design

The INTERGROWTH-21st Project principally aimed to evaluate growth, health, nutrition, and development from less than 14 weeks' gestation to 2 years of age, using the conceptual framework of WHO Multicentre Growth Reference Study (MGRS) [16]. For the current analysis, we used blood pressures that were taken as part of the study protocol.

**Setting.** The INTERGROWTH-21st Project was carried out from 2009 to 2016 across 8 diverse geographically delimited urban areas: Pelotas (Brazil), Turin (Italy), Muscat (Oman), Oxford (United Kingdom), Seattle (United States of America), Shunyi County in Beijing (China), central Nagpur (India), and the Parklands suburb of Nairobi (Kenya). Area and hospital selection has been previously described [13]. Women receiving antenatal care had to plan to deliver in these institutions or in a similar hospital located in the same geographical area.

**Participants.** Participants were selected based upon WHO criteria for optimal health, nutrition, education, and socioeconomic status needed to construct international standards [13]. Women aged $\geq$18 and <35 years old with a body mass index (BMI) <30 kg/m$^2$ and height $\geq$153 cm, at low risk of adverse maternal and perinatal outcomes, who commenced antenatal care before 14 weeks' gestation with reliable menstrual dates, and a confirmatory ultrasound dating scan and provided written informed consent are the FGLS population. Exclusion criteria included hypertension (defined as systolic $\geq$140 mmHg or diastolic $\geq$90 mmHg) in a previous pregnancy or the first trimester of the present pregnancy; chronic hypertension on treatment; and a past history of preeclampsia, eclampsia, or haemolysis, elevated liver enzymes and low platelets (HELLP) syndrome.

FGLS also excluded women whose pregnancies became complicated by a priori specified criteria including fetal death, congenital abnormality, severe or catastrophic medical morbidity not evident at enrolment (such as cancer or HIV), severe unanticipated conditions related to the pregnancy that required admission to hospital (such as severe preeclampsia or eclampsia), and those identified during the study who no longer fulfilled entry criteria (such as women who started smoking or had a malaria episode) [13].

**Bias.** Entry criteria for FGLS were chosen to balance the strict WHO-recommended criteria for selecting a healthy population with external validity of the results [13].

**Variables, data sources, and measurement.** Gestational age was calculated from the last menstrual period (LMP) date, provided it was certain; the woman had a regular 24- to 32-day menstrual cycle; she had not been using hormonal contraception or breastfeeding in the preceding 2 months; and any discrepancy between the gestational ages based on LMP and crown rump length (CRL), measured by ultrasound between 9 and 13$^{+6}$ weeks' gestation, was $\leq$7

days. The ultrasound dating scan was undertaken using standard study criteria for measuring CRL [13].

The instruction manual for measurement techniques, methods for multicentre standardisation of measures, and procedures for equipment calibration and maintenance is published [17]. All documentation, protocols, data collection forms, and electronic transfer strategies are available at www.intergrowth21.org. Trained research personnel measured systolic and diastolic blood pressure using an automated machine validated in pregnancy (Microlife Blood Pressure Monitor for Pregnant Women, Microlife USA, Florida, USA) with an appropriately sized cuff on study entry between 9 and $13^{+6}$ weeks' gestation and every 5 ± 1 weeks until delivery (i.e., from enrolment, blood pressure was measured in the gestational age windows 14 to 18, 19 to 23, 24 to 28, 29 to 33, 34 to 38, and 39 to 42 weeks).

**Study size.**   The sample size for FGLS was based on practical and statistical considerations [18]. FGLS established an average sample of 500 pregnant women per study site, after exclusion of complicated pregnancies (approximately 3%) and those lost to follow-up (estimated to be 3%, S1 Text). This sample size was adequate to explore site-specific differences [18].

**Quantitative variables.**   We included all recorded blood pressures from all participants in the main analysis and constructed smoothed centiles for systolic and diastolic blood pressure by gestational age. We modelled blood pressure at fortnightly gestational age windows from 12 to 40 weeks. We constructed international, gestational age-specific, smoothed centiles for blood pressure following WHO recommendations [19].

**Statistical methods.**   We included a statistical analysis plan in our application to use data from the INTERGROWTH-21st Project (S2 Text). We assessed variation in systolic and diastolic blood pressure between sites [18] to explore whether we could pool data. We used analysis of variance (ANOVA) to calculate the percentage of variance in the longitudinal blood pressure measures from variance between sites adjusted for gestational age (fixed effects). We treated sites and individuals as random effects. We calculated a standardised site difference (SSD, similar to a z score) as the difference between the mean of 1 site and the mean of all sites. We expressed differences as a proportion of the all sites' standard deviation (SD) at each corresponding gestational age. The SSD allows direct comparisons across gestational age windows. A priori, we specified an overall value of £ 0.5 SSD as adequate for combining data from all sites [18], as described previously for the construction of international standards in WHO MGRS and INTERGROWTH-21st studies [14]. We undertook a post hoc analysis of the effect of removing outlying sites (with SSDs >0.5).

We estimated blood pressure centiles from generalised additive models for location, scale, and shape (GAMLSS) framework. We assessed different distributions for both systolic and diastolic blood pressure within the GAMLSS framework. This included Box-Cox Cole and Green, Box-Cox Power Exponential, Box-Cox-t, Skew Power Exponential type 3, Skew t type 3, and Power Exponential. We used penalised splines and fractional polynomials to create smooth centiles across the gestational age range. We chose the best fitting distribution based on model fit (Akaike information criterion and Bayesian information criterion) and a comparison of fitted versus empirical centiles. We chose the same distribution for all subgroups within any given blood pressure.

We estimated the precision of the centiles via bootstrapping, by repeatedly sampling and analysing the dataset 50 times. We used the SD of those bootstrapped estimates to calculate the 95% confidence interval for each centile at 2-week intervals. We used the R (version 3·4; R Foundation, Vienna, Austria, www.r-project.org) and GAMLSS (version 4·3–3; R Foundation, www.gamlss.com) packages for all analyses.

We conducted a post hoc sensitivity analysis to assess the effect of excluding women who developed hypertension (defined as systolic ≥140 mmHg or diastolic ≥90 mmHg at any

antenatal visit), constructing gestational age-specific centiles for blood pressure in women who remained normotensive (blood pressure <140/90 mmHg).

Finally, we performed post hoc analyses to explore changes in systolic and diastolic blood pressure by baseline for all women and by blood pressure quartile. Our statistical analysis plan included further subgroup analyses (booking BMI, booking BP, and maternal age), which will be considered in future publications.

**Loss to follow-up and missing data.** Where a woman did not contribute blood pressure measures within one of the possible gestational age windows (14 to 18, 19 to 23, 24 to 28, 29 to 33, 34 to 38, and 39 to 42 weeks), we included all those available. Where women did not complete the study but we knew the final outcome, we included all the data available unless consent for data use was withdrawn.

## Ethical approval

The INTERGROWTH-21st Project was approved by Oxfordshire Research Ethics Committee "C" (reference: 08/H0606/139), research ethics committees of the individual institutions, and relevant regional health authorities.

## Results

### Participants

The enrolment strategy and eligibility criteria of the INTERGROWTH-21st Project, at population and individual level, are published [13]. In brief, 13,108 pregnant women were screened at <14 weeks' gestation. Of these, 4,607 (35%) met eligibility criteria, provided written informed consent, and were enrolled. Common exclusion reasons were maternal height <153 cm (1,022/8,501; 12%), BMI $^3$30 kg/m$^2$ (1,009/8,501; 12%), and age <18 or >35 years (915/8,501; 11%) at screening. During pregnancy, 71 women (2%) were lost to follow-up or withdrew consent. A total of 36 were excluded (29 had severe medical conditions, 6 took up smoking, and 1 used recreational drugs). Moreover, 4,422 women delivered a live-born singleton. Of these, 4,321 (98%) had a baby without a congenital malformation, (Fig 1), the same cohort that contributed to the INTERGROWTH-21st Fetal Growth Standards [14].

### Descriptive data

The published sociodemographic characteristics [14] were similar across sites (Table A in S3 Text). The mean maternal age was 28.4 (SD 3.9) years; 97% (4,204/4,321) of women were married or living with a partner, and 68% (2,955/4,321) were nulliparous. Their mean BMI was 23.3 (SD 3.0) kg/m$^2$. The median gestational age at the first antenatal visit was 11.8 (SD 1.4) weeks.

### Maternal and perinatal outcome data

Maternal and perinatal outcome data have been published [20]. In brief, 132 (3%) developed gestational hypertension and 31 (<1%) preeclampsia. The spontaneous initiation of labour, preterm birth, term low birth weight, cesarean section, and neonatal mortality rates were 66.4%, 4.5%, 3.0%, 35.7%, and 0.2%, respectively.

### Main results

Blood pressure was measured a median of 6 (range = 1 to 7) times throughout pregnancy, resulting in 25,027 blood pressure measures. Within-site systolic and diastolic blood pressure variation (36.6% and 38.4%) was around 6 times higher than between sites variation (6.3% and 6.6%). The all sites' SD for systolic and diastolic blood pressure ranged from 10.3 and 7.6

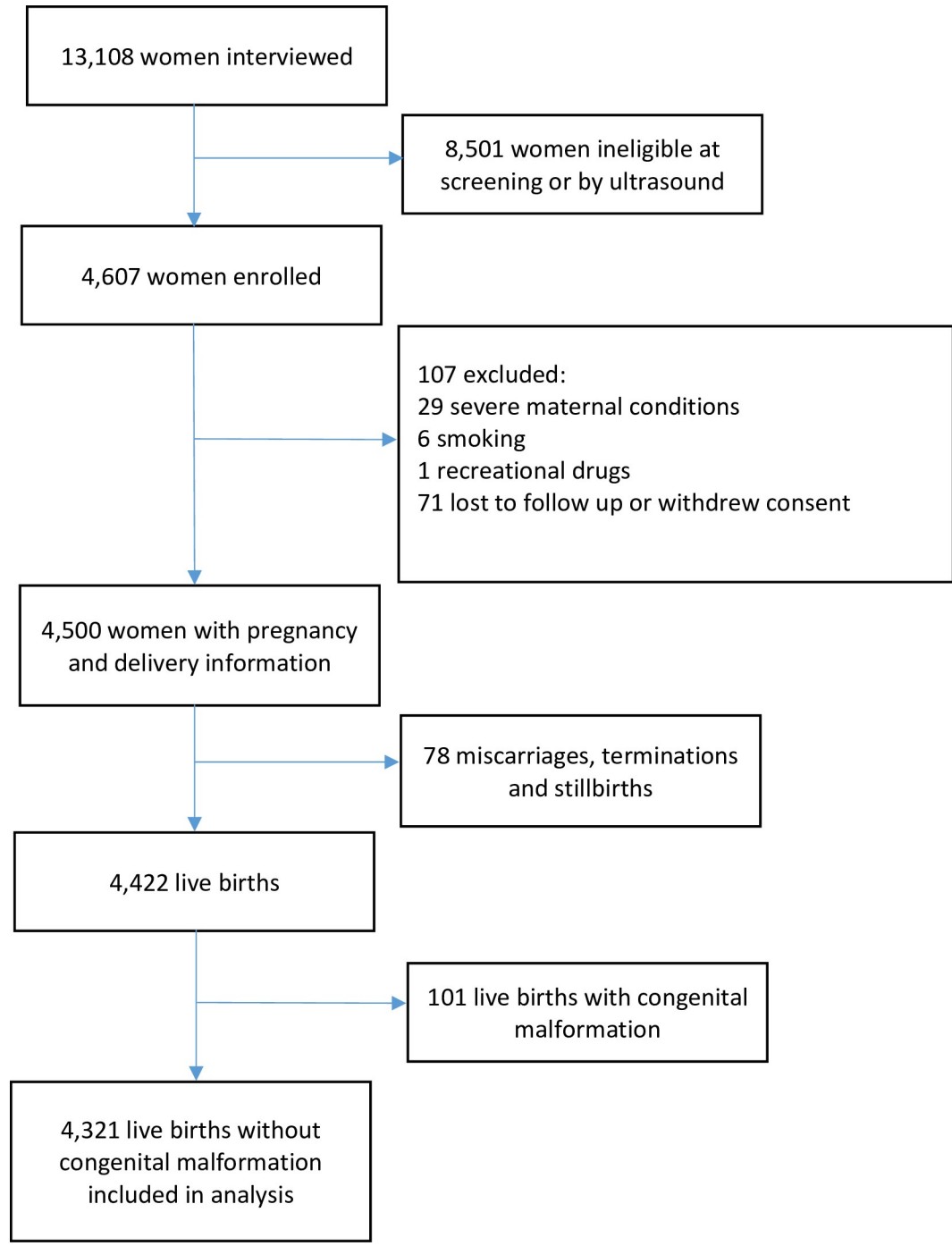

**Fig 1. Flow diagram illustrating women enrolled in FGLS.** FGLS, Fetal Growth Longitudinal Study.

mmHg, respectively, at 9 to $13^{+6}$ weeks to 11.6 and 8.5 mmHg, respectively, at 34 to $40^{+6}$ weeks' gestation.

Within 6 gestational age windows from 9 to $40^{+6}$ weeks, representing 48 comparisons per blood pressure, 39 (systolic) and 42 (diastolic) had SSDs <0.5 (as prescribed a priori in the INTERGROWTH-21st Study Protocol [18]) of the SD of all sites combined (Figs 2 and 3, Table B in S3 Text). Of the 9 comparisons of systolic blood pressure that were ≥0.5 SSD, the

sites in India and the UK contributed 5 and 4, respectively, although the difference was <0.5 for both sites at 9 to 13$^{+6}$ weeks' gestation (−0·12 and 0.33, respectively; Fig 2). Of the 6 comparisons of diastolic blood pressure that were >0.5 SSD, sites in Italy and the UK contributed 5 and 1, respectively, although the difference was <0.5 at 9 to 13$^{+6}$ weeks' gestation (Fig 3). We investigated the effect of removing these potential outlying sites from the dataset used to construct the centiles: The effect was minimal, so all were included (Fig A and Table B in S3 Text).

For all women, the median systolic blood pressure was lowest at 12 weeks' gestation: 111.5 (95% CI 111.3 to 111.8) mmHg rising to a maximum of 119.6 (95% CI 118.9 to 120.3) mmHg at 40 weeks' gestation, a difference of 8.1 (95% CI 7.4 to 8.8) mmHg. Diastolic blood pressure decreased from 12 weeks' gestation: median 69.1 (95% CI 68.9 to 69.3) mmHg to 68.5 (95% CI 68.3 to 68.7) mmHg at 19$^{+5}$ weeks' gestation, a change of −0.6 (95% CI −0.8 to −0.4) mmHg. Diastolic blood pressure then increased to a maximum of median 76.3 (95% CI 75.9 to 76.8) mmHg at 40 weeks' gestation, a minimum (at 19$^{+5}$ weeks) to maximum difference of 7.8 (95% CI 7.3 to 8.2) mmHg.

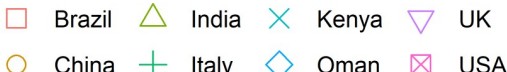

**Fig 2. SSD for systolic blood pressure in the FGLS of the INTERGROWTH-21st Project.** FGLS, Fetal Growth Longitudinal Study; SBP, systolic blood pressure; SD, standard deviation; SSD, standardised site difference.

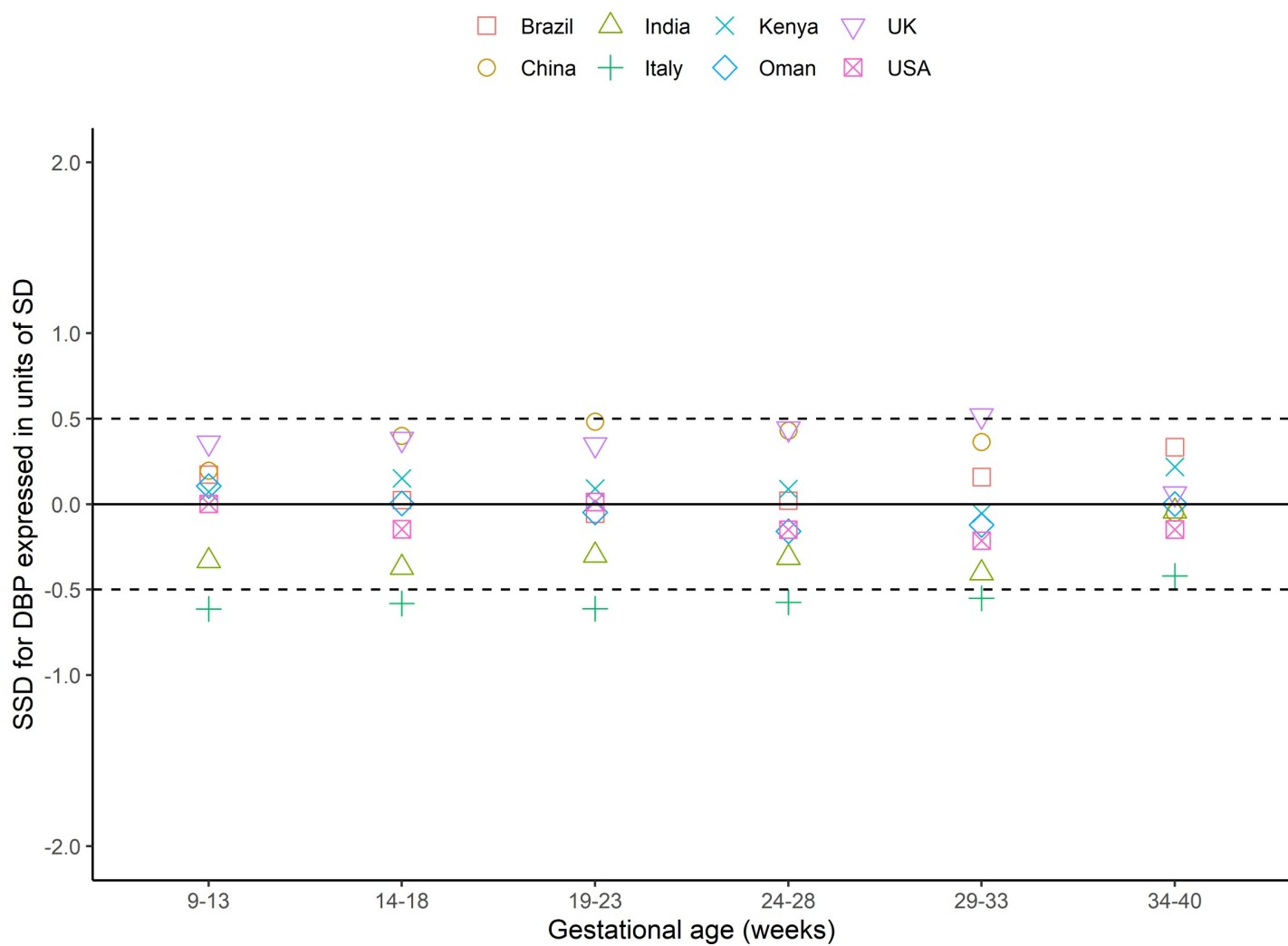

**Fig 3. SSD for diastolic blood pressure in the FGLS of the INTERGROWTH-21st Project.** DBP, diastolic blood pressure; FGLS, Fetal Growth Longitudinal Study; SD, standard deviation; SSD, standardised site difference.

Fig 4 represents the smoothed, pregnancy-specific, third, 10th, 50th, 90th, and 97th centiles for systolic and diastolic blood pressure. Gestational age-specific values for the smoothed centiles and a plot of the smoothed centiles with associated 95% CI can be found in Table C and Fig B in S3 Text.

## Other analyses

A sensitivity analysis of normotensive pregnancies (excluding women who developed hypertension, defined as systolic ≥140 mmHg or diastolic ≥90 mmHg, at any follow-up visit) showed similar blood pressure patterns (Fig C in S3 Text).

Fig 5 shows the absolute change from baseline blood pressure for all women. Systolic blood pressure fell by >14 mmHg or diastolic blood pressure by >11 mmHg from baseline in fewer than 10% of women at any gestational age. Systolic blood pressure fell by >16 mmHg or diastolic blood pressure by >11 mmHg in only 3% of women at 40 weeks' gestation. Under 10% of women increased their systolic blood pressure by >24 mmHg or diastolic blood pressure by

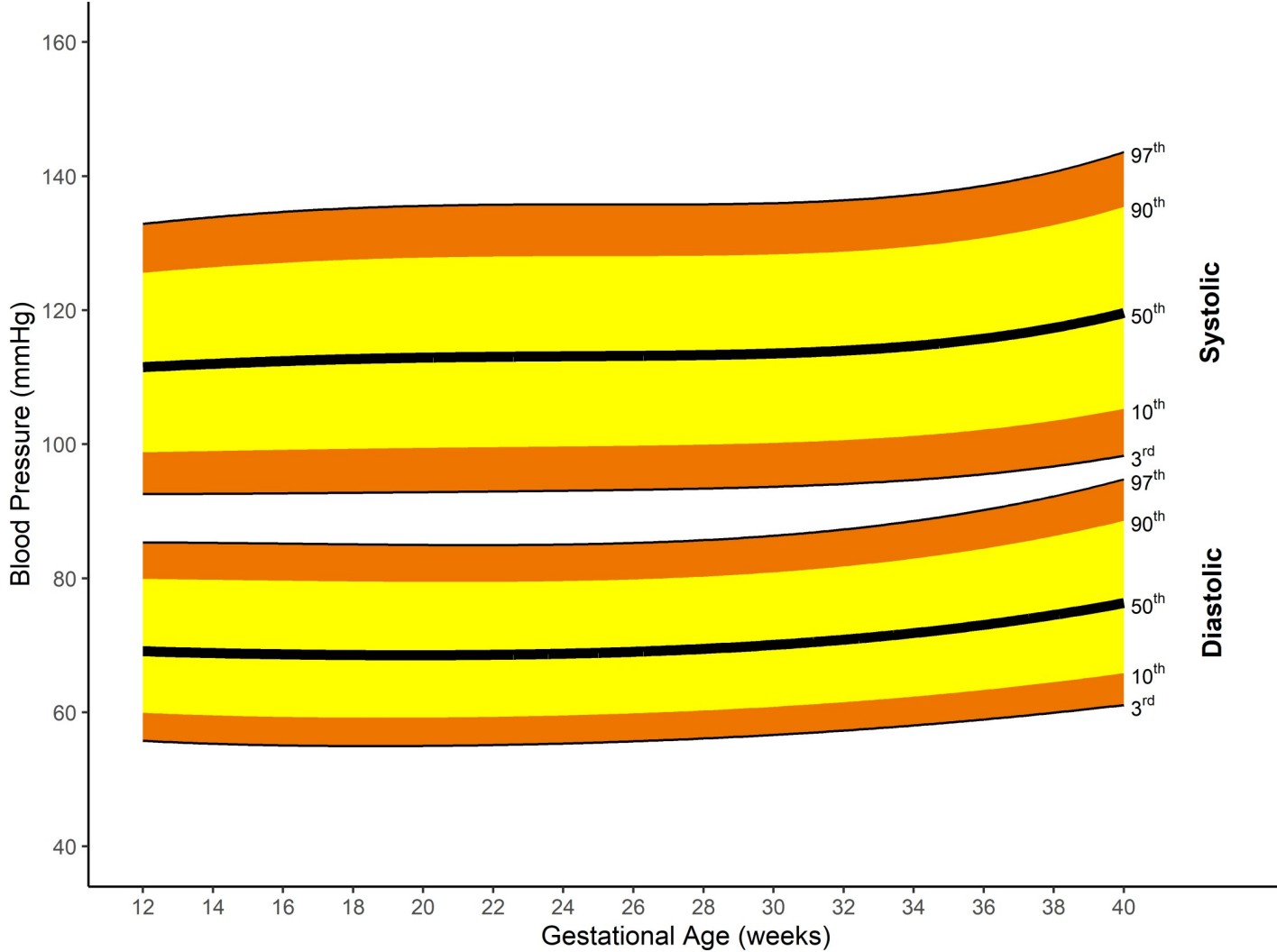

**Fig 4. Smoothed centiles for systolic and diastolic blood pressure in mmHg; third, 10th, 50th, 90th, and 97th centiles.**

>18 mmHg at any gestational age. Less than 3% of women increased their systolic blood pressure by >34 mmHg or diastolic blood pressure by >26 mmHg.

Smoothed centiles for blood pressure by gestational age, and change from baseline blood pressure, depending on quartile of baseline blood pressure at study enrolment are presented (Figs D and E in S3 Text). The figures demonstrate a strong regression to the mean effect.

## Discussion

Our population-based study includes over 4,000 healthy women from 8 diverse countries, which allowed construction of international, evidence-based, gestational age-specific centiles for blood pressure throughout pregnancy. Systolic blood pressure rose through pregnancy, with no mid-second trimester drop. The diastolic blood pressure nadir at $19^{+5}$ weeks' gestation was only 0.6 mmHg lower than measures taken at study entry <14 weeks' gestation. A decrease of >14 mmHg from the baseline systolic or diastolic blood pressure was unusual (<10% of all women), particularly in the latter stages of pregnancy. The average increase in term blood pressure from baseline was <10 mmHg for both systolic and diastolic measures.

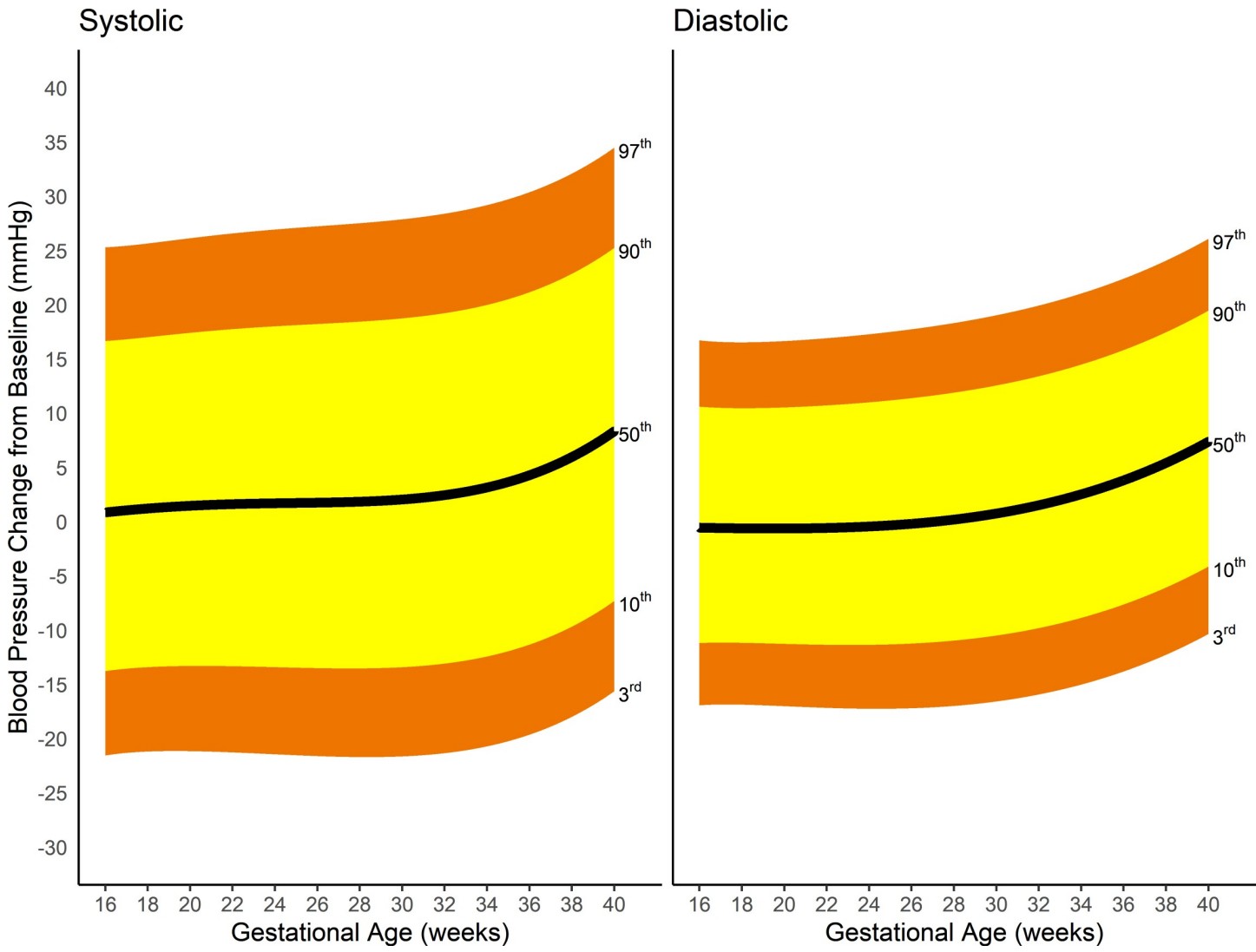

**Fig 5. Changes from baseline for both systolic and diastolic blood pressure from 16 weeks' gestation for third, 10th, 50th, 90th, and 97th centiles.**

Systolic blood pressure fell by >14 mmHg or diastolic blood pressure by >11 mmHg from baseline in fewer than 10% of women at any gestational age. Fewer than 10% of women increased their systolic blood pressure by >24 mmHg or diastolic blood pressure by >18 mmHg at any gestational age.

## Interpretation

Our study allows clinicians to interpret blood pressure in pregnancy in the light of international gestational age-specific centiles. It shows that the initial blood pressure in pregnancy can be used to facilitate interpretation of subsequent measures. These findings should aid clinicians in recognising when a pregnant woman is deviating from "normal" values. The proportion of total variance attributed to population differences between sites was <7%, and removing potentially outlying sites did not alter our findings, supporting the position that population-specific ranges for blood pressure in pregnancy are not required.

## Comparison with other studies

**Blood pressure.** Our findings are contrary to much published literature upholding a clinically significant mid-second trimester drop. Data underlying this view are often dated [9,10], routinely collected [21,22], single centre [9,10], and based on non-standardised measurement of blood pressure [9,10] or devices not ratified for use in pregnancy [10,21–23]. Larger, routinely acquired birth cohorts from single geographical regions show smaller drops [12,24]. More recent, prospective, standardised studies refute this dogma [25,26]. Our systematic review, including over 36,000 women, found an almost identical progressive rise in systolic blood pressure and approximately 1 mmHg mid-second trimester drop in diastolic blood pressure [27]. Taken together, clinically significant drops in population blood pressure from booking to delivery clearly do not occur in modern practice.

Confusion may have arisen because studies use different definitions of baseline blood pressure. Where prepregnancy blood pressure (rather than the first blood pressure taken in pregnancy) is used as the baseline, a small drop is generally seen mid-second trimester [11,23]. Although physiologically interesting, recent prepregnancy blood pressures are not usually available. Blood pressure patterns during pregnancy, as revealed in our study, are probably more important clinically.

Women in the lowest quartile of booking blood pressure demonstrated the largest rise in blood pressure, while the highest quartile had the greatest range of likely decreases, showing regression to the mean. These prospective findings add to previous work from routinely acquired data [12].

## Strengths and limitations of study

To our knowledge, this is the first study to collect blood pressure data from across the world using the prescriptive approach recommended by WHO for construction of international standards. This is also the largest prospective study using an automated method of blood pressure measurement with a machine validated in pregnancy.

By adopting these approaches to collect data prospectively from healthy women in a population-based study involving 8 geographically diverse sites, we are confident that the international centiles are both robust and representative of women of optimal health, nutrition, education, and socioeconomic status.

**Limitations.** The study has limitations. The lack of prepregnancy blood pressure data may disguise a relative drop in blood pressure in early pregnancy. However, as a prepregnancy measure is rarely available in routine clinical practice, comparison with blood pressure in early pregnancy contributes to clinical applicability.

We did not exclude the few women who developed gestational hypertension (3%) or pre-eclampsia (<1%), rather, we chose to demonstrate that excluding these women did not affect our findings (Fig C in S3 Text). The large size of the study made it impractical to define the time of day at which blood pressure was measured to allow exploration of circadian effects. In our study, blood pressure was not measured in duplicate; however, this mirrors clinical practice. The INTERGROWTH-21st Project recruited women with a BMI $<30$ kg/m$^2$; therefore, the applicability of the findings to women with a BMI $\geq 30$ kg/m$^2$ is uncertain. Further work is needed to determine how knowledge of these centiles affects the detection of deteriorating health in pregnant women.

## Policy implications

There is an indisputable need for pregnancy-specific early warning scores that incorporate blood pressure changes [5]. Current national definitions of severe maternal sepsis either do

not include hypotension as a contributory factor (UK Obstetric Surveillance System [2]) or stipulate a physiologically extreme drop in systolic blood pressure >40 mmHg (UK Sepsis Trust [28]). We show that it is highly abnormal for a healthy woman to demonstrate a systolic or diastolic blood pressure drop of half this magnitude, making a strong case for utilising less severe thresholds for systolic hypotension in maternal sepsis definitions. Current MOEWS systems are derived from expert opinion [29], but could be better evidenced using blood pressure centile thresholds, an approach shown to be effective in nonpregnant adults [30].

## Conclusions

We present international, gestational age-specific centiles for blood pressure based on a healthy population of low-risk women with good pregnancy outcomes. We show clear limits for acceptable change in blood pressure during pregnancy, which should help clinicians determine patients with abnormal blood pressure rises and falls. Our findings challenge the frequently quoted mid-second trimester drop in blood pressure, advocating for a higher index of clinical suspicion when a woman presents with a "lower than booking" blood pressure, especially in late pregnancy. These gestational age-specific centiles should help build an adaptive, intelligent, evidence-based MOEWS to allow earlier recognition of the unwell pregnant woman.

## Transparency declaration

PW guarantees that the manuscript is an honest, accurate, and transparent account of the study being reported and that no aspects have been omitted.

## Supporting information

**S1 STROBE Checklist. STROBE, Strengthening the Reporting of Observational Studies in Epidemiology.**
(DOC)

**S1 Text. Sample size calculation.**
(DOCX)

**S2 Text. Statistical analysis plan.**
(XLSX)

**S3 Text. Supporting information. Table A in S3 Text:** Baseline characteristics for women enrolled in the FGLS. **Table B in S3 Text:** All sites and individual site means (SD) for DBP and SBP of all women. **Fig A in S3 Text:** Plot to illustrate sensitivity analysis for excluding potential site outliers; SBP and DBP according to gestational age for third, 50th, and 97th centiles. **Table C in S3 Text:** Smoothed centiles for SBP and DBP according to gestational age for third, 10th, 50th, 90th, and 97th centiles (95% CI). **Fig B in S3 Text:** Smoothed centiles for SBP and DBP in mmHg; third, 10th, 50th, 90th, and 97th centiles with corresponding 95% confidence intervals. **Fig C in S3 Text:** Smoothed centiles for SBP and DBP in mmHg excluding women ($n$ = 132) who developed hypertension (systolic BP ≥140 or diastolic BP ≥90); third, 10th, 50th, 90th, and 97th centiles. **Fig D in S3 Text:** SBP and DBP from 16 weeks' gestation onwards for quartiles according to baseline blood pressure (third, 10th, 50th, 90th, and 97th centiles). First quartile 76–105/40–64 mmHg; second quartile 106–111/65–70 mmHg; third quartile 112–119/71–75 mmHg; fourth quartile ≥120/76 mmHg). **Fig E in S3 Text:** Change in systolic and DBP during pregnancy by quartiles of baseline blood pressure at study entry at 9–13$^{+6}$ weeks' gestation. DBP, diastolic blood pressure; FGLS, Fetal Growth

Longitudinal Study; SBP, systolic blood pressure; SD, standard deviation; SSD, standardised site difference.
(DOCX)

**S4 Text. Contributions to membership of the INTERGROWTH-21st and its Committees.**
INTERGROWTH-21st, International Fetal and Newborn Growth Consortium for the 21st Century.
(DOCX)

## Acknowledgments

We would like to thank the health authorities in Pelotas, Brazil; Beijing, China; Nagpur, India; Turin, Italy; Nairobi, Kenya; Muscat, Oman; Oxford, United Kingdom; and Seattle, United States of America who facilitated the project by allowing participation of these study sites as collaborating centres. We are extremely grateful to Philips Medical Systems who provided the ultrasound equipment and technical assistance throughout the project. We also thank MedSci-Net U.K. for setting up the INTERGROWTH-21st website and for the development, maintenance, and support of the online data management system.

We thank the parents and infants who participated in the studies and the more than 200 members of the research teams who made the implementation of this project possible. The participating hospitals included Brazil, Pelotas (Hospital Miguel Piltcher, Hospital São Francisco de Paula, Santa Casa de Misericórdia de Pelotas, and Hospital Escola da Universidade Federal de Pelotas); China, Beijing (Beijing Obstetrics & Gynecology Hospital, Shunyi Maternal & Child Health Centre, and Shunyi General Hospital); India, Nagpur (Ketkar Hospital, Avanti Institute of Cardiology Private Limited, Avantika Hospital, Gurukrupa Maternity Hospital, Mulik Hospital & Research Centre, Nandlok Hospital, Om Women's Hospital, Renuka Hospital & Maternity Home, Saboo Hospital, Brajmonhan Taori Memorial Hospital, and Somani Nursing Home); Kenya, Nairobi (Aga Khan University Hospital, MP Shah Hospital and Avenue Hospital); Italy, Turin (Ospedale Infantile Regina Margherita Sant' Anna and Azienda Ospedaliera Ordine Mauriziano); Oman, Muscat (Khoula Hospital, Royal Hospital, Wattayah Obstetrics & Gynaecology Poly Clinic, Wattayah Health Centre, Ruwi Health Centre, Al-Ghoubra Health Centre and Al-Khuwair Health Centre); UK, Oxford (John Radcliffe Hospital); and USA, Seattle (University of Washington Hospital, Swedish Hospital, and Providence Everett Hospital).

Author contributions to Membership of International Fetal and Newborn Growth Consortium for the 21st Century (INTERGROWTH-21st) and its committees are available in S4 Text.

Full acknowledgement of all those who contributed to the development of the INTERGROWTH-21st Project protocol appears at www.intergrowth21.org.uk.

## Author Contributions

**Conceptualization:** Stephen H. Kennedy, J. Alison Noble, José Villar.

**Data curation:** Stephen Gerry, Eleonora Staines Urias, Cesar Victora, Eric Ohuma.

**Formal analysis:** Lauren J. Green, Lucy Mackillop, Stephen Gerry, Peter Watkinson.

**Funding acquisition:** Stephen H. Kennedy.

**Investigation:** Manorama Purwar, Fernando Barros, Aris T. Papageorghiou.

**Methodology:** J. Alison Noble.

**Project administration:** Manorama Purwar, Leila Cheikh Ismail, Fernando Barros, Maria Carvalho, Yasmin Jaffer, Michael Gravett, Ruyan Pang, Ann Lambert, Enrico Bertino, Aris T. Papageorghiou, Cutberto Garza, Zulfiqar Bhutta, José Villar.

**Supervision:** Stephen H. Kennedy, Manorama Purwar, Leila Cheikh Ismail, Fernando Barros, Maria Carvalho, Yasmin Jaffer, Michael Gravett, Ruyan Pang, Ann Lambert, Enrico Bertino, Aris T. Papageorghiou, Cutberto Garza, Zulfiqar Bhutta, José Villar, Peter Watkinson.

**Validation:** Aris T. Papageorghiou.

**Visualization:** Stephen H. Kennedy, José Villar, Peter Watkinson.

**Writing – original draft:** Lauren J. Green, Stephen H. Kennedy, Lucy Mackillop, Stephen Gerry, Peter Watkinson.

**Writing – review & editing:** Manorama Purwar, Eleonora Staines Urias, Leila Cheikh Ismail, Fernando Barros, Maria Carvalho, Eric Ohuma, Yasmin Jaffer, J. Alison Noble, Michael Gravett, Ruyan Pang, Ann Lambert, Enrico Bertino, Aris T. Papageorghiou, Cutberto Garza, Zulfiqar Bhutta, José Villar.

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
