## [Editor Report · Decision Letter 0]

21 Dec 2020

Dear Dr Green, 

Thank you for submitting your manuscript entitled "International gestational age-specific centiles for blood pressure in pregnancy from the INTERGROWTH-21st Project: a prospective longitudinal cohort study in eight countries" for consideration by PLOS Medicine.

Your manuscript has now been evaluated by the PLOS Medicine editorial staff as well as by an academic editor with relevant expertise and I am writing to let you know that we would like to send your submission out for external assessment.

Once your full submission is complete, your paper will undergo a series of checks in preparation for external assessment. Once your manuscript has passed all checks it will be sent out for assessment. 

Kind regards,

Richard Turner, PhD

rturner@plos.org

---

## [Decision Letter · Decision Letter 1]

20 Feb 2021

Dear Dr. Green,

Thank you very much for submitting your manuscript "International gestational age-specific centiles for blood pressure in pregnancy from the INTERGROWTH-21st Project: a prospective longitudinal cohort study in eight countries" (PMEDICINE-D-20-06147R1) for consideration at PLOS Medicine. 

Your paper was evaluated by an academic editor with relevant expertise and sent to independent reviewers, including a statistical reviewer. The reviews are appended at the bottom of this email and any accompanying reviewer attachments can be seen via the link below:

[LINK]

In light of these reviews, we will not be able to accept the manuscript for publication in the journal in its current form, but we would like to invite you to submit a revised version that addresses the reviewers' and editors' comments fully. You will appreciate that we cannot make a decision about publication until we have seen the revised manuscript and your response, and we expect to seek re-review by one or more of the reviewers. 

We hope to receive your revised manuscript by Mar 15 2021 11:59PM. Please email us (plosmedicine@plos.org) if you have any questions or concerns.

Please let me know if you have any questions, and we look forward to receiving your revised manuscript in due course. 

Sincerely,

Richard Turner, PhD

rturner@plos.org

Please adapt the study descriptor in your title to refer to the present analysis rather than the parent study. If this analysis is a retrospective analysis of prospectively-gathered data, please remove the word "prospective" from the title. 

Please combine the "Methods" and "Results" subsections of your abstract. The final sentence of the new combined subsection should begin "Study limitations include ..." or similar, and should summarize 2-3 of the study's main limitations. 

Please quote some additional information on participant demographics in the abstract. 

Please trim the "Conclusions" subsection of your abstract, which should summarize the study's conclusions (claims about "the first" and "the largest" can be removed). 

After the abstract, please remove the "Research in context" information and substitute a new and accessible "Author summary" section in non-identical prose. You may find it helpful to consult one or two recent research papers in PLOS Medicine to get a sense of the preferred style.

Please indicate in the Methods section whether the present analysis had a protocol or prespecified analysis plan, and if so attach the document as a supplementary file, referred to in the text. Please highlight analyses that were not prespecified. 

Please restructure the start of the Discussion section so that the first paragraph provides a summary of the study's main findings. 

Throughout the text, please adapt reference call-outs as follows: "... mid-trimester [11,23].".

Please remove information on funding and competing interests from the title page and from the end of the main text. In the event of publication, this information will appear in the article metadata, via entries in the submission form. 

Please ensure that all references contain full access information, e.g., reference 3. 

Please supply a completed STROBE checklist with your revision as a supplementary file, labelled "S1_STROBE_Checklist" or similar and referred to as such in your methods section. In the checklist, please refer to individual items by section (e.g., "Methods") and paragraph number rather than by line or page numbers, as the latter generally change in the event of publication. 

Comments from the reviewers:

*** Reviewer #1: 

This report presents the secondary analysis of a prospective, longitudinal, observational cohort study (2009-16) across eight diverse urban areas in Brazil, China, India, Italy, Kenya, Oman, United Kingdom, and United States.

The stated primary aim of the analysis was to describe systolic and diastolic cohort blood pressure patterns in healthy pregnancy. The authors have conducted sensitivity analyses to assess the effect of excluding women who developed hypertension (defined as systolic ≥140 mmHg or diastolic ≥90 mmHg at any follow-up visit), and constructed international gestational age-specific centiles for blood pressure in women who remained normotensive (blood pressure <140/90 mmHg). 

Comments:

This study is reported to follow STROBE guidelines. Can the authors please supply a copy of their STROBE checklist in the supplementary material?

Given the extent of the relevant supporting documentation, in terms of protocols, data collection forms, and electronic transfer strategies, it is perhaps appropriate that the authors site where readers can access these (www.intergrowth21.org), rather than include copies within the supplementary material.

"Statistical considerations focused on the precision and accuracy of a single centile and regression based reference limits. FGLS established an average sample of 500 pregnant women per study site, after exclusion of complicated pregnancies (approximately 3%) and those lost to follow-up (estimated to be 3%). This sample size was adequate to explore site-specific differences (18). "

Can more detail on these sample size calculations please be provided by the authors in brief within the supplementary material?

"We constructed International, gestational age-specific, smoothed centiles for blood pressure, i.e. 3rd, 10th, 50th, 90th and 97th centiles with corresponding 95% confidence intervals, following WHO recommendations (19). "

The analytical approach applied is robust and rigorous, with a summary of techniques provided in the Methods section. 

The statistical methodology is technically appropriate and recommended practice for the data type, research question and study context in hand, including that applied for the subgroup and post hoc analyses.

*** Reviewer #2: 

I appreciated reviewing this manuscript, which represents an important topic of interest in prenatal/antenatal care. The INTERGROWTH-21st Study has brought very critical evidence to obstetrics, and this manuscript promises to continue this trend. However, I have some concerns about it in its current form.

1. Throughout the paper, you've used the term "mid-trimester". It may be a case of minor linguistic differences, but to me this term means "in the middle of any trimester" and not in the middle of a specific trimester. At one point you do use "mid-second trimester" and it would be clearer to an international audience to use this phrasing each time.

2. The Research in Context section is meant for general audiences, but is currently very jargony. I think that the Introduction and Discussion of the paper use more accessible language that could be paraphrased here.

3. In the Introduction, the MOEWS needs to be briefly described where it is introduced.

4. The Methods section is in need of reorganization. All of the subsections that are about the participants (Participants, Bias, Study size) should be together, and all of the sections that are about the analyses (Variables, Quantitative variables, [neither of which, I would argue, describe variables], Other analyses, and Statistical methods) should be together. After rearranging, the grouped subsections will need to be edited to remove redundancy and to more clearly describe the flow of your analysis plan as it is presented in the Results and Supplements.

5. The figure designs need improvement. The symbols representing the Standardized Site Differences in Figure 2 are difficult to discern and need to be enlarged. Also in Figure 2, the SBP plot should go before the DBP plot to be consistent with the title and the presentation of the findings. In many of the remaining figures, SBP and DBP are not clearly distinguished from each other. The color scheme for most of the figures (3, 4, S3-5.2) is not particularly aesthetic, and it is not clear to me why color blocks were used instead of lines.

*** Reviewer #3: 

Reviewer comments:

Abstract:

The authors have performed a unique analysis generating gestational age-specific centiles using blood pressure data from a population-based study including women from across the world. Their results are important as they provide information on international standards. Moreover, these results provide insight into physiological blood pressure changes in pregnancy. Overall, this is a clearly designed and very well-written study and the fact that it is prospective adds value to the data. The study uses strong statistical methods and analyses. 

I only do have some minor comments:

Minor:

- Abstract: 

Results section: Systolic blood pressure: …. , difference (95% CI) 8.1 (7.4 to 8.8) mmHg. This sentence is unclear. At least add the word "a" as is written in the results section. However, the authors preferably state more clearly only a difference in systolic blood pressure of 8.1 mmHg (95% CI: 7.4-8.8) between 12 weeks' gestation and 40 weeks' gestation is found. 

- Description results: 

I suggest to adapt the description of the results to: Median (then value) and subsequent (95% CI: values) which is easier to read and then understand the results.

- Conclusions: 

 The conclusion that these data provide a strong basis for constructing… (line) is not supported by these data. Hence, in the abstract the authors should use describe it as accordingly in the main paper-discussion: (Line 480-482): These gestational age-specific centiles should help build an….

- 

Results:

Line 355: minimum to maximum difference. The authors should also refer to and add the gestational age then: is it second to third trimester? 

Discussion:

Interpretation:

Line 408-409: These findings should aid clinicians in recognizing when a pregnant woman is becoming unwell. 

I disagree with this sentence as these findings as for now only help to recognize deviations from normal values. 

The authors should change this interpretation of their data as they have not investigated that their values aid in recognizing when a pregnant woman is becoming unwell. 

Furthermore, the phrase becoming unwell is too vague. They should state more specifically what they mean by this word.

Additionally, they should state more clearly that it still needs to be thoroughly investigated these values are of additional value for the detection of clinically ill women. Then only, subsequently, as they mention in their conclusion the results will help build an MOEWS to allow earlier recognition. 

Limitations: 

With the aim of this study to describe blood pressure patterns in a healthy pregnancy women (with an increased risk of ) developing complications are excluded. The authors succeeded as for example a low incidence of gestational hypertension and preeclampsia is reported. They furthermore demonstrated in a sensitivity analysis that exclusion of these women did not affect their results.

However, they should mention in their limitations that the strict inclusion criteria (e.g age 18-35 years, BMI < 30) as described in their methods may contribute to less generalizability of their gestational age-specific centiles for blood pressure. Specifically, taking their future conclusion into account as their results should help clinicians determine patients with abnormal blood pressure rise and falls. 

Limitations:

The authors indeed mention that lack of pre-pregnany data may disguise first-trimester changes. They however then jump to the following conclusion (line 453-454): comparison with blood pressure in early pregnancy has greater clinical applicability. 

I think this statement is overrated and this conclusion cannot be drawn as there was no comparison with pre-pregnancy data as these are lacking.

Hence, suggest to change this sentence to: in routine clinical practice, blood pressure in early pregnancy contributes to clinical applicability.

***

[LINK]

---

## [Editor Report · Decision Letter 2]

24 Mar 2021

Dear Dr. Green,

Thank you very much for re-submitting your manuscript "International gestational age-specific centiles for blood pressure in pregnancy from the INTERGROWTH-21st Project: a longitudinal cohort study in eight countries" (PMEDICINE-D-20-06147R2) for consideration at PLOS Medicine.

I have discussed the paper with editorial colleagues and our academic editor and I am pleased to tell you that, provided the remaining editorial and production issues are dealt with, we expect to be able to accept the paper for publication in the journal.

[LINK]

In revising the manuscript for further consideration here, please ensure you address the specific points made by the editors. In your rebuttal letter you should indicate your response to the editors' comments and the changes you have made in the manuscript. Please submit a clean version of the paper as the main article file. A version with changes marked must also be uploaded as a marked up manuscript file.

Please let me know if you have any questions in the meantime, and we look forward to receiving the revised manuscript shortly.   

Sincerely,

Richard Turner, PhD

rturner@plos.org

Requests from Editors:

The current version of the paper has a notable level of text matching with a published paper (Ohuma et al) on the INTERGROWTH project, mainly in the Methods section. We ask you to reword the relevant sections of the text to reduce this. 

Please make arrangements for data deposition in a suitable publicly-accessible repository, and update the data statement in your paper. 

In the title, please move "in 8 countries" before the colon. 

Please remove "only" at line 66.

At line 68 and any other instances, we suggest rephrasing "... dropped their blood pressure" (to, e.g., "Systolic blood pressure fell in fewer ..."). 

Please bullet the points in the Author summary, aiming for 3 subsections of 3 points each, comprising 1-2 short sentences.

If participants' provision of informed consent is not mentioned in the Methods section, please add this. 

Please adapt the label for the STROBE checklist to "S1_STROBE_Checklist" or similar and refer to it by this label in the text.

Throughout the text, please use the style "8 diverse countries", except at the start of sentences (e.g., "Eight diverse countries ..."). 

Please remove the information on copyright from the end of the main text (the paper will be published under a CC-BY licence). 

Please remove the information on the INTERGROWTH groups from the end of the main text (this can be transferred to a Supplementary file, if you wish). 

***

---

## [Editor Report · Decision Letter 3]

3 Apr 2021

Dear Dr Green, 

On behalf of my colleagues and the Academic Editor, Prof Persson, I am pleased to inform you that we have agreed to publish your manuscript "International gestational age-specific centiles for blood pressure in pregnancy from the INTERGROWTH-21st Project in 8 countries: a longitudinal cohort study" (PMEDICINE-D-20-06147R3) in PLOS Medicine.

PRESS

Sincerely, 

Richard Turner, PhD 

rturner@plos.org